# *Pediococcus pentosaceus* OL77 Enhances Oat (*Avena sativa*) Silage Fermentation Under Cold Conditions

**DOI:** 10.3390/microorganisms13102248

**Published:** 2025-09-25

**Authors:** Xin Liu, Guiqin Zhao, Jie Bai, Xinyi Qu, Jikuan Chai, Doudou Lin

**Affiliations:** College of Pratacultural Science, Gansu Agricultural University, Lanzhou 730070, China; liux9688@163.com (X.L.); baij@gsau.edu.cn (J.B.); 17794291920@163.com (X.Q.); chaijk@gsau.edu.cn (J.C.); lindd1014@163.com (D.L.)

**Keywords:** bacterial community, lactic acid bacteria, low-temperature ensiling, microbial ecology, silage quality

## Abstract

Ensiling forage under low-temperature conditions often leads to poor fermentation and nutrient losses. This study evaluated the effects of a cold-tolerant *Pediococcus pentosaceus* OL77 strain on oat silage. Silages were prepared with or without *Pediococcus pentosaceus* inoculation (1 × 10^5^ cfu/g FM). After 90 days, OL77-treated silage showed markedly higher lactic acid (45.83 vs. 30.51 g/kg DM), lower pH (3.88 vs. 4.443), and better preservation of WSC (64.68 vs. 47.60 g/kg DM) and crude protein (89.26 vs. 65.52 g/kg DM) than the control. Microbial analysis revealed accelerated colonization by *Pediococcus*, reduced bacterial diversity, and faster stabilization of the fermentation process. Functional predictions indicated enhanced carbohydrate and energy metabolism. These findings demonstrate that OL77 can effectively improve fermentation quality and nutrient preservation of oat silage under low-temperature conditions, offering a practical inoculant option for cold regions.

## 1. Introduction

The Qinghai–Tibetan Plateau is one of the major pastoral regions in China; however, forage production in this area faces severe challenges due to its high altitude and cold climate. The short growing season of forage and the limited availability of natural grassland resources have led to a pronounced shortage of forage. Unlike many regions where scarcity mainly occurs in autumn and winter, on the Qinghai–Tibetan Plateau the prolonged cold season and delayed onset of pasture regrowth result in forage shortage extending into spring [1]. Against this background, oat, owing to its high yield, superior quality, and strong adaptability to low-temperature environments, has been widely grown in cold regions and has become one of the primary forage sources on the Qinghai–Tibet Plateau [2,3]. However, its harvest often coincides with frequent rainfall in autumn, causing substantial difficulties in hay production [4]. Under these circumstances, making silage is a more effective and convenient approach [5,6]. Nevertheless, constrained by the local climatic conditions, silage production still faces technical challenges in temperature control, fermentation quality and storage management, especially low temperature during ensiling in autumn and winter, which limited fermentation process and effectiveness [7,8].

Silage preservation relies on the fermentation of lactic acid bacteria (LAB), which converts soluble carbohydrates into organic acids, which have the ability to inhibit the growth of spoilage microorganisms, thus ensuring long-term storage of the forage [1,9]. However, low-temperature conditions limited the metabolic activity of LAB, slowing down the fermentation process and leading to a slower reduction in pH and an increased risk of nutrient loss [9]. The inoculation of LAB in silage could effectively improve fermentation success. However, traditional commercial inoculants have difficulties adapting to the harsh and cold environmental conditions of the Tibetan Plateau [10]. Our team previously found that the inoculation of commercial LAB improved oat silage fermentation to some extent in the Gannan region. Inoculated silages exhibited slightly lower pH values and higher lactic acid concentrations compared with the untreated control. However, even under inoculation, fermentation still required about 90 days to complete, and the extended process resulted in considerable dry matter and crude protein losses, thereby reducing overall silage quality [11]. These results highlight the limitations of commonly used commercial inoculants under cold conditions.

To address this problem, it is necessary to explore LAB strains that are tolerant to low temperatures and capable of rapidly initiating fermentation. Previous studies have shown that microorganisms in extreme environments gradually develop unique structural and physiological adaptations that enhance survival [12,13]. The unique ecological and climatic conditions of the Tibetan Plateau have fostered distinct LAB resources, and strains that have persisted in this environment are likely to possess enhanced adaptability to cold and harsh conditions [14].

The microbial community compositions and functional characteristics of silage undergo dynamic changes during the fermentation process. These changes not only affect silage quality but are also closely related to the fermentation traits of silage forage and the nutritional value of the feed [15]. PacBio sequencing combined with SMRT (Single Molecule Real-Time) technology has been used to track changes in microbial communities at species level in silage, as it allows for the bacterial characterization of target samples at the species level. Compared to traditional high-throughput sequencing methods, PacBio sequencing can accurately capture the structural information of microbial genomes through long reads, avoiding common issues with assembly errors or genome assembly difficulties seen with short-read technologies [16]. Recent research has expanded beyond the species composition of microbial communities to focus more on the functional prediction of microorganisms [17,18]. Functional analysis of microbial communities helps to understand the ecological roles played by microorganisms in specific environments. Through functional prediction, researchers can not only reveal changes in microbial community diversity but also explore the intrinsic relationship between microbial communities and silage quality [19].

Previous studies have shown that *Pediococcus* species can rapidly initiate silage fermentation, suggesting potential for improving silage preservation under cold conditions [20,21]. In our preliminary work, we isolated a cold-tolerant strain, *Pediococcus pentosaceus* OL77, from oats collected on the Qinghai–Tibetan Plateau, which was able to grow at 5–10 °C. We hypothesized that inoculation with *P. pentosaceus* OL77 would accelerate fermentation, enhance lactic acid production, and improve the overall fermentation quality of oat silage at low temperature by shaping a more favorable microbial community. Therefore, this study aimed to evaluate the effects of *P. pentosaceus* OL77 on the fermentation characteristics and microbial community dynamics of oat silage under cold conditions.

## 2. Materials and Methods

### 2.1. Preparation of Oat Silage Feed

The silage material used in the experiment was oat variety Longyan 3 grown in Dachigou Town, Tianzhu Tibetan Autonomous County, Gansu Province (E102°15′, N36°45′; elevation 2594 m). The average temperature during the oat silage completion period in this region is below 10 °C. The forage was harvested at milk stage on 28 August 2023, and chopped into 2–3 cm lengths using a silage chopper (9ZP-0.8, Qufu Shengjia Heavy Industry Co., Ltd., Qufu, Shandong, China). The LAB inoculant used, *Pediococcus pentosaceus* OL77, was isolated and selected by our research team in previous studies and has been patented (Patent number: 202110128172.7). It is currently preserved at the China Center for Type Culture Collection (CCTCC, No. M2020290). The traditional commercial *Pediococcus pentosaceus* (TCP) was purchased from Vita Plus Corporation (Lallemand Inc., Montréal, Québec, Canada).

The oat at the milk stage was chopped and air-dried in a cool, ventilated place until the dry matter content reached 346.24 ± 4.85 g/kg FM. The fresh forage had a pH of 6.27, soluble carbohydrates content of 141.15 ± 6.07 g/kg DM, crude protein content of 103.4 ± 2.44 g/kg DM, neutral detergent fiber (NDF) content of 528.36 ± 9.65 g/kg DM, and acid detergent fiber (ADF) content of 331.42 ± 7.05 g/kg DM. The populations of LAB, aerobic bacteria, molds, and yeasts on the surface of the oats were 4.32 ± 0.03, 6.06 ± 0.05, 2.85 ± 0.02, and 3.76 ± 0.03 log10 CFU/g FM, respectively.

The experiment included three additive treatments: a control group (CK) with no additives, a low-temperature *Pediococcus pentosaceus* (OL77) treatment group, and a traditional commercial *Pediococcus pentosaceus* (TCP) treatment group. The freshly cultured bacterial suspension in the logarithmic growth phase was used at an inoculation rate of 1 × 10^5^ cfu/g FM. During silage preparation, the bacterial inoculants were sprayed onto the forage as a 20 mL/kg FM solution and thoroughly mixed. The CK received the same volume of sterile water. For each treatment, 500 g of forage was sealed in polyethylene plastic bags (250 × 350 mm), vacuum packed, and stored at local ambient temperature for fermentation. Samples were taken at 3, 7, 14, 60, and 90 d, with three replicates at each time point.

### 2.2. Determination of Chemical and Nutritional Parameters

A 20 g fresh sample was mixed with 180 mL of sterile water in a juicer and blended at high speed for 30 s. The mixture was first filtered through cheesecloth and then through filter paper to obtain the filtrate, which was immediately measured for pH using a pH meter (PHS-3C, Shanghai Youke Instrument Co., Ltd., Shanghai, China). The filtrate was then divided into two portions. One portion was used to measure organic acids, including lactic acid, acetic acid, propionic acid, and butyric acid, while the other portion was used for ammonia nitrogen determination and stored at −20 °C. One portion of the filtrate was acidified to approximately pH 2.0 with 7.14 mol/L H_2_SO_4_ (7.14 M) for organic acid analysis. The acidified filtrate was then filtered through a 0.22 μm filter. The contents of lactic acid (LA), acetic acid (AA), propionic acid (PA), and butyric acid (BA) were determined using high-performance liquid chromatography (HPLC: Agilent 1260; column: Shodex Rspak KC-811 S-DVB gel, 30 mm × 8 mm; mobile phase: 3 mmol/L perchloric acid; column temperature: 50 °C; detector: DAD detector; Injection volume: 5 μL; Flow rate: 1 mL/min; detection wavelength: 210 nm). The second portion of the filtrate was used to measure ammonia nitrogen content. A 40 mL unacidified filtrate was mixed with 10 mL of 25% trichloroacetic acid (TCA, *w*/*v*) at a 4:1 ratio and allowed to settle at 4 °C overnight to precipitate proteins. After high-speed centrifugation (10,000 r/min, 4 °C) for 15 min, the supernatant was used to measure ammonia nitrogen content using the phenol-hypochlorite method [22,23].

The determination of dry matter (DM) was performed using a two-step procedure to account for volatile compound loss. A 200 g sample was placed in a kraft paper bag and pre-dried in a 65 °C oven for approximately 72 h. The pre-dried samples were then ground to pass through a 40-mesh screen (0.425 mm aperture). The final DM content was subsequently determined by drying a sub-sample to a constant weight at 105 °C (AOAC, 2005; method 934.01) [24]. All results for chemical analysis were expressed on this final, corrected DM basis. The ground samples were analyzed for crude protein (CP), neutral detergent fiber (NDF), acid detergent fiber (ADF), and water-soluble carbohydrates (WSC). Total nitrogen (TN) content was determined using an automatic Kjeldahl nitrogen analyzer (K9840, Hannon Instruments Ltd., Jinan, China), and crude protein (CP) content was calculated by multiplying TN by 6.25 [25]. The contents of NDF and ADF were assessed according to the method described by Van Soest [26]. For WSC determination, 100 mg of dry powder was weighed into a test tube, and 15 mL of distilled water was added. The mixture was boiled for digestion, then cooled and filtered to obtain the filtrate. Anthrone reagent was added to the filtrate, boiled again, then cooled, and the optical density was measured at a wavelength of 620 nm. The WSC content in the sample was calculated using a standard curve [27].

### 2.3. Bacterial Community Detection

A 0.5 g silage sample from each treatment was placed in a 1.5 mL extraction tube for genomic DNA extraction. The concentration and purity of the extracted DNA were determined using a UV spectrophotometer (NanoDrop 2000, Thermo Fisher Scientific, Waltham, MA, USA), and integrity was verified by 1% agarose gel electrophoresis. Only DNA samples meeting the required quality and quantity standards were used for downstream analyses. The full-length bacterial 16S rRNA gene was amplified by PCR with the forward primer 27F (5′-AGRGTTYGATYMTGGCTCAG-3′) and reverse primer 1492R (5′-RGYTACCTTGTTACGACTT-3′). Sample-specific 16 bp barcodes were incorporated into the primers for multiplexing during single-molecule real-time (SMRT) sequencing. PCR amplification was performed under the following conditions: initial denaturation at 95 °C for 30 s, annealing at 57 °C for 30 s, extension at 72 °C for 60 s, and a final extension at 72 °C for 5 min. PCR products were purified using Agencourt AMPure Beads (Beckman Coulter, Indianapolis, IN, USA) and quantified with the PicoGreen dsDNA Assay Kit (Invitrogen, Carlsbad, CA, USA). After Qubit quantification, purified PCR products were pooled in equimolar concentrations according to sequencing requirements. Library preparation was conducted with the SMRTbell Template Prep Kit 1.0-SPv3 (PacBio, Menlo Park, CA, USA), and sequencing was performed on the PacBio platform using the DNA/Polymerase Binding Kit 3.0. (PacBio, Menlo Park, CA, USA). Circular Consensus Sequencing (CCS) reads were generated, filtered, clustered, and denoised. Taxonomic annotation and abundance profiling were subsequently performed. Alpha and beta diversity indices, correlation analyses, and functional predictions were carried out as described previously. Specifically, the Shannon index for alpha diversity was calculated based on operational taxonomic unit (OTU) abundance using the QIIME pipeline, and beta diversity was evaluated using principal coordinates analysis (PCoA) based on Bray–Curtis distance matrices [15]. Using PICRUSt2 software (v2.5.1), the potential functions of bacterial communities at different fermentation stages of oat silage were predicted. Redundancy analysis (RDA) was employed to investigate the relationship between the fermentation characteristics of oat silage and the bacterial community composition.

### 2.4. Statistical Analysis

Data for chemical composition, fermentation quality, and microbial counts were analyzed using the General Linear Model (GLM) procedure in SPSS (version 26.0, Inc., Chicago, IL, USA). A two-way analysis of variance (ANOVA) was applied to determine the effects of the treatments. The statistical model included the fixed effects of silage additive, silage temperature, and their interaction (additive × temperature). Before analysis, the normality of the residuals was assessed using the Shapiro–Wilk test. When a significant effect was detected by the ANOVA, means were separated using Tukey’s honestly significant difference (HSD) post hoc test. Statistical significance was declared at *p* < 0.05, and a trend was considered when 0.05 ≤ *p* < 0.10. Data are presented as means with the standard error of the mean (SEM).

## 3. Results

### 3.1. Fermentation Parameters of Oat Silage with Microbial Inoculants at Low Temperature

The effects of microbial inoculants on fermentation parameters of oat silage at low temperature are shown in Table 1. The results indicated that pH, lactic acid concentration, acetic acid concentration, propionic acid concentration, butyric acid concentration, and the LA/AA ratio were all significantly influenced by fermentation time (D), additive treatment (T), and their interaction (D × T) (*p* < 0.001). The pH value decreased more rapidly in the OL77 group when compared with other groups, in which the pH decreased rapidly to 5.84 after 3 d of ensiling. The OL77 group maintained the lowest pH throughout the silage phase, followed by the TCP group, and the pH value of the CK was higher than the OL77 and TCP groups. The differences in lactic acid concentration among the three groups was consistent with the difference in pH, that is, the lactic acid concentration of the OL77 group was the highest during the whole ensiling periods, followed by the TCP group, and the lactic acid concentration of the CK was the lowest. The OL77 group had the lowest acetic acid content (13.85 g/kg DM) compared to the TCP and CK groups after 90 d of ensiling. However, at days 3, 7, and 14 of fermentation, the OL77 group had the highest acetic acid content. The TCP and OL77 groups exhibit significantly higher propionic acid content during the fermentation period than the CK group. Butyric acid was not detected in the OL77 group during the entire silage period, while small amounts were detected in the TCP and CK groups at 60 and 90 d of fermentation. The LA/AA ratio in the OL77 group showed an initial increase followed by a decline throughout the fermentation period, reaching its highest value of 4.19 at day 14. The LA/AA ratio in the TCP and CK groups was significantly lower than that in the OL77 group during the whole ensiling periods.

### 3.2. Effect of Pediococcus pentosaceus on Chemical Composition of Oat Silage at Low Temperature

The effects of *Pediococcus pentosaceus* on the nutritional composition of oat silage are shown in Table 2. The results of variance analysis indicated that the interaction between ensiling days and additives had no significant effect on the dry matter content of low-temperature oat silage. Additives also had no significant effect on DM and WSC content. The dry matter content in the OL77 and TCP groups was significantly higher than that in the CK group after 90 d of ensiling. In the OL77 group, the lowest WSC content was observed in the OL77 group when compared with the TCP and CK after 3, 7, and 14 of ensiling, while at the later stages of fermentation (60 and 90 d), the WSC content in the OL77 group was significantly higher than in the CK and TCP groups (*p* < 0.05). The CP content decreased in all groups, with the CK group showing a more pronounced decrease throughout the fermentation period, from 102.88 g/kg DM to 78.52 g/kg DM. In contrast, the decline in the CP content was less pronounced in the OL77 and TCP groups, especially in the OL77 group, where the CP content remained at 89.26 g/kg DM even after 90 d of fermentation. Both NDF content and ADF content decreased gradually during fermentation, with the degradation being more significant in the inoculated groups, especially in the OL77 group. Ammonia nitrogen (NH_3_-N), a byproduct of protein degradation, significantly increased during fermentation. The NH_3_-N content in the CK group reached its highest level at 90 d, while the OL77 and TCP groups showed significantly lower NH_3_-N content than the CK group (*p* < 0.05).

### 3.3. Bacterial Community Diversity of Oat Silage Inoculated with Pediococcus pentosaceus at Low Temperature

As shown in Figure 1A, there were no significant differences in the Shannon index among the three treatment groups after 3 d of ensiling, while a lower Shannon index was observed in the OL77 group compared with the TCP and CK groups from 7 to 90 d of ensiling. In addition, the Shannon index was lower in the TCP group than in the CK group from 60 to 90 d of ensiling. Figure 1B shows the beta diversity of the bacterial community of oat silage using principal coordinates analysis (PCoA). The CK group mainly distributed in the lower half of the PCoA2 axis, while the OL77 and TCP groups were more concentrated in the upper half. From a temporal perspective (different colors), microbial community structure showed dynamic changes at different time points. The samples from all groups were widely scattered, and no clear clustering trends were observed after 3 and 7 d of ensiling. After 14 d of ensiling, the samples from CK, OL77, and TCP groups began to show distinct distribution trends, especially the separation of OL77 and TCP groups in the PCoA space. In the later stages of fermentation (60 d and 90 d), the microbial community structures separated significantly. The CK group clustered in the upper-right region of the PCoA plot, while OL77 and TCP samples clustered in the lower-left and upper-left regions, respectively, forming independent clusters. Notably, the OL77 group samples clustered in the lower-left region, showing a significant difference from the other treatment groups. TCP samples clustered in the upper-left region, indicating a certain stability in the community but still showing clear separation from both OL77 and CK groups.

### 3.4. Bacterial Community Compositions of Oat Silage Inoculated with Pediococcus pentosaceus at Low Temperature

At the genus level (Figure 2A), the main bacteria attached to fresh oat samples before silage were *Weissella*, *Pantoea*, *Hafnia*, *Pseudomonas*, and *Lactiplantibacillus*. During the ensiling period, the abundance of *Weissella* decreased in all groups. Notably, the relative abundance of *Pediococcus* in the OL77 inoculated group was higher than that in the TCP group after 3 days of silage. As the ensiling time progressed, the *Pediococcus* abundance remained highest in the OL77 group when compared with TCP and CK. Throughout the ensiling period, the highest relative abundance of *Pseudomonas* and *Enterobacter* were observed in the CK compared to the OL77 and TCP groups, and the TCP inoculant group also exhibited some abundance of *Pseudomonas* and *Enterobacter*. However, in the OL77 inoculant group, the relative abundance of *Pseudomonas* and *Enterobacter* was extremely low, with *Enterobacter* not detected by 90 d.

The bacterial community composition and dynamics at the species level during the oat silage process are shown in Figure 2B. The relative abundance of *Pediococcus pentosaceus* in fresh oat samples was very low, at only 2.30%. After the ensiling, the relative abundance of *Pediococcus pentosaceus* remained low in the CK, while *Levilactobacillus brevis* and *Lactiplantibacillus plantarum* gradually increased and became dominant. Meanwhile, *Enterobacter asburiae* and *Enterococcus gallinarum* were observed in the CK during the entire ensiling period. In the TCP group, the relative abundance of *Pediococcus pentosaceus* increased slowly during the early ensiling period (3 and 7 d), reaching only 38.65% at day 14. However, *Pediococcus pentosaceus* increased rapidly after 7 d of ensiling and became the dominant bacteria during the 14 to 90 d of ensiling in the OL77 inoculated group. The relative abundance of *Hafnia alvei* in the OL77 group decreased throughout the ensiling process, while it showed a trend of first increasing and then decreasing in the TCP and CK. *Enterobacter asburiae* was present in the TCP group during the whole ensiling process.

### 3.5. Bacterial Community Functions Oat Silage

The predicted KEGG pathways were classified into five major functional modules: cellular processes, environmental information processing, genetic information processing, metabolism, and organismal systems, which were further subdivided into several sub-modules (Figure 3A).

The clustering heatmap (Figure 3B) shows the dynamic changes in the main bacterial functions during the fermentation period of oat silage. Overall, the microbial functional activities in the OL77 and TCP treatment groups were significantly different from the CK, especially in the later stages of fermentation. In the early fermentation stages (3 d and 7 d), carbohydrate metabolism, energy metabolism, and amino acid metabolism in the TCP treatment group were shown as light blue, while these metabolic activities were significantly stronger in the OL77 group. By the mid-fermentation stage (14 d), the majority of functional categories in the OL77 group exhibited more prominent red regions, especially in carbohydrate metabolism, amino acid metabolism, and energy metabolism, where OL77 showed more activity than TCP. Additionally, in the OL77 group, functions such as signal transduction, cell growth, and cell death were showed higher abundances. After 60 d of ensiling, most of the metabolic functions in the OL77 treatment group reached a high activity level, reflected by significant red regions, particularly in carbohydrate metabolism, amino acid metabolism, and energy metabolism. After 90 d of ensiling, the metabolic functions in the OL77 treatment group had overall decreased, with the heatmap color transitioning toward light blue. In contrast, the TCP treatment group showed more pronounced changes in metabolic activity at 60 d and 90 d, indicating that the fermentation process in the TCP silage was slower, and the fermentation completion was not as high as in the OL77 treatment group.

### 3.6. Correlation Analysis Between Bacterial Communities and Fermentation Characteristics in Oat Silage

Figure 4 illustrates the relationship between the fermentation characteristics of oat silage and the bacterial community composition. The relatively long arrows represented WSC content, lactic acid concentration, and pH. *Lactiplantiba cillus plantarum* and *Levilactobacillus brevis* form acute angles with the vectors for LA and Aconcentration when connected to the origin, demonstrating a positive correlation between these bacteria and the fermentation traits. *Pediococcus pentosaceus* is positioned along the extended vectors of WSC and LA. In contrast, these LAB species exhibit negative correlations with pH, ammonia nitrogen (NH_3_-N), neutral detergent fiber (NDF), and acid detergent fiber (ADF). The reduction in pH and NH_3_-N levels reflects increased acidity and the inhibition of protein degradation during fermentation. Additionally, decreases in NDF and ADF suggest enhanced digestibility of the silage, which is an optimal outcome of the fermentation process. Species such as *Weissella koreensis* and *Enterococcus gallinarum* are located near the vectors for NDF, ADF, and pH, implying their involvement in fiber degradation and the facilitation of feed digestibility. However, excessive growth of these bacteria within the silage environment may inhibit lactic acid production, impede the rapid decline of pH, and consequently compromise the quality and stability of the silage. Furthermore, the presence of bacterial communities positively correlated with NH_3_-N, including *Hafnia alvei* and *Enterobacter asburiae*, may be associated with protein degradation and ammonia nitrogen production. It is essential to monitor and control the activity of these bacteria during fermentation to prevent adverse effects on the normal metabolic functions of LABs.

## 4. Discussion

### 4.1. Nutritional and Fermentative Quality of Oat Silage Under Low-Temperature Ensiling

Numerous studies have demonstrated that excessively low temperatures impair the natural ensiling process by delaying acidification and promoting undesirable microbial proliferation [28,29]. This study found that the pH value significantly decreased during the early stages of fermentation after inoculation with OL77. This rapid acidification reflects the strong adaptability of OL77 to low-temperature environmental conditions, and also indicates that *Pediococcus pentosaceus* is the most effective fermentation starter for silage in the early stages, which is consistent with the study by Liu et al [30].

During the entire fermentation process, the lactic acid concentration of OL77-inoculated silage remained higher than that of TCP silage. This observation is consistent with findings in previous studies [10,30], where strains isolated from environments with distinct ecological conditions demonstrated enhanced adaptability and superior fermentation performance. This may be due to OL77 being isolated from oat materials in the high-altitude cold environment of the Tibetan Plateau, which may enhance its adaptation to low-temperature silage fermentation. This is in agreement with the report by Wang et al. (2023), who suggested that even within the same species of lactic acid bacteria, ecological adaptability differences based on their origin may significantly affect their performance and effectiveness during silage fermentation [31]. The concentration of propionic acid, an antifungal agent, was also found to be higher in OL77 silage during the later stages of fermentation, suggesting an enhanced antifungal capability of the silage. The production of propionic acid may be linked to secondary microbial activity, which contributes to the preservation of silage by inhibiting fungal growth [32]. LA/AA is an indicator for evaluating the quality of silage fermentation and measuring the type of fermentation, with an ideal range of 2.5–3.5 [33]. The higher the ratio, the more likely it is that lactic acid fermentation is dominant and the fermentation quality is higher. If the ratio is too low, there may be abnormal fermentation, leading to a decrease in silage quality [34]. In this study, inoculation with OL77 consistently maintained a high LA/AA ratio throughout the entire fermentation process, indicating that lactic acid fermentation was predominant and ensuring good fermentation quality.

DM is fundamental for assessing the nutritional composition of silage, as its content directly influences the fermentation quality of the silage. After 90 days of fermentation, the final DM content in the OL77- and TCP-inoculated groups was significantly higher than that in the control group. A comparison with the initial DM values showed that the DM loss in these treatment groups was significantly lower than in the control group. This indicates that the addition of lactic acid bacteria can effectively reduce DM loss. This finding was consistent with the results of Sun et al. [35]. Ellis et al. [36] reported that the rapid decrease in pH during the early stages of ensiling is crucial for inhibiting the growth of undesirable microorganisms and reducing nutrient loss in silage. Chen et al. [19] found that the WSC content added with lactic acid bacteria was significantly increased in all fermentation periods compared with the CK. In this study, the WSC content of OL77 inoculated silage decreased in the early stage of fermentation compared with the CK, but increased significantly in the later stage. This could be attributed to the reduced activity of most microorganisms in the low-temperature environment. After the addition of low-temperature-resistant lactic acid bacteria OL77, more WSC was consumed during the early stage to support the rapid growth and establishment of the dominant microbial flora. The significant increase in WSC content in the later stages was likely due to the more complete fermentation in the OL77 inoculated silage, resulting in reduced WSC consumption [20].

Studies have shown that the addition of effective LAB reduces CP loss and further decreases the formation of NH_3_-N [30,37]. In the present study, OL77 inoculation enhanced CP retention and reduced NH_3_-N accumulation relative to CK, indicating that OL77 effectively inhibited the growth of undesirable microorganisms and reduced protein degradation. The OL77 inoculation also decreased NDF and ADF contents, which could be attributed to the potential indirect promotion of fiber degradation by OL77. This could occur through the stimulation of fibrolytic microorganisms or by facilitating acid hydrolysis. Relevant studies have suggested similar mechanisms, indicating that OL77 may enhance fiber degradation through microbial interactions or chemical processes [35,38]. However, the precise mechanisms responsible for these effects warrant further investigation, particularly at the enzymatic or strain-specific level.

### 4.2. Effect of Pediococcus pentosaceus OL77 on the Microbial Community Structure of Oat Silage Fermentation at Low Temperature

Inoculation with *Pediococcus pentosaceus* OL77 significantly shaped the microbial ecology of oat silage under low-temperature conditions. Alpha diversity indices were markedly reduced in OL77 and TCP-treated groups compared to the uninoculated control, consistent with previous findings that LAB dominance leads to ecological narrowing during early fermentation [12]. This reduction in species richness is likely driven by rapid pH decline, which selectively inhibits acid-sensitive taxa [38], enabling the early establishment of lactic acid-dominant microbiota. The results of the PCoA showed that fermentation time significantly influenced the microbial community structure of oat silage, with more pronounced differences in community structure between treatment groups observed at 60 and 90 d. This finding contrasts with the report by Xu et al. [16], who found no significant change in bacterial beta diversity between LAB-inoculated and CK silage. This discrepancy may be attributed to the fact that in this study, the OL77 and TCP treatments formed more stable and significantly different microbial communities in the later stages of fermentation, while the CK’s microbial community structure primarily relied on the natural competition and adaptation of environmental microorganisms, leading to weaker community stability.

Throughout the entire ensiling process, the relative abundance of *Pediococcus pentosaceus* in the OL77 group remained consistently higher than in both the TCP and CK groups. Since both OL77 and TCP were inoculated with *Pediococcus pentosaceus* additives, this indicates that OL77 can adapt to the low-temperature fermentation environment and proliferate during ensiling. In the present study, *Enterobacter*, *Pseudomonas*, and *Enterococcus* were identified from fresh oat before ensiling. *Pseudomonas* is widely distributed in nature and can produce bioactive substances, resist various plant diseases, and is also a known pathogen which can cause infections in humans and animals [17,33]. During the fermentation process, *Enterobacter* species not only produce small quantities of organic acids but also generate undesirable fermentation byproducts, such as ammonia nitrogen and other amines. These byproducts contribute to poor silage odor and a decline in fermentation quality [39]. In the present study, the relative abundance of *Enterobacter asburiae* and *Enterococcus gallinarum* in the CK increased progressively as the fermentation process advanced. Meanwhile, the presence of *Enterobacter asburiae* was observed throughout the entire fermentation period in the TCP-inoculated group. However, *Enterobacter asburiae* was only detected in the OL77 inoculated silage during the early stages of ensiling. The results indicated that OL77 inoculation effectively inhibited the growth of undesirable microorganisms, thereby improving the fermentation quality. Hafnia, a genus in the family *Enterobacteriaceae*, has proteolytic activity and can deaminate and decarboxylate certain amino acids [40]. In this study, Hafnia exhibited higher relative abundance in both the TCP and CK, and protein content decreased compared to the OL77 group during the early ensiling period. This suggests that Hafnia may play a role in the proteolysis of oat proteins during the early fermentation phase, a result consistent with Bai et al. [15]. Similarly, *Pantoea* is also considered an undesirable bacterium during ensiling because it competes with LAB for substrates. In addition, some *Pantoea* species may produce toxins that pose potential health risks to animals [13]. In this study, the abundances of Hafnia and *Pantoea* significantly decreased over the fermentation period, likely due to growth inhibition by low pH in the OL77 treatment group. This reduction not only minimized the loss of nutritional components in silage oats but also ensured the quality and safety of the silage. After 90 d of ensiling, higher concentrations of acetic acid were detected in the CK, which might be produced by *Levilactobacillus brevis*, which is a heterofermentative lactic acid bacterium [41]. In addition, *Pediococcus pentosaceus* was almost entirely dominated in the OL77 inoculated silage after 60 and 90 d of ensiling. This indicated that the microbial community reached a stable ecological phase, where a few acid-tolerant species occupied the ecological niches, thus forming a more stable microbial community structure [33]. Collectively, these results demonstrate that OL77 inoculation not only accelerates fermentation kinetics but also rewires microbial succession trajectories to favor functionally beneficial taxa, thereby improving silage preservation under cold stress.

### 4.3. Effect of Pediococcus pentosaceus OL77 on the Bacterial Functional Profile of Oat Silage Fermented at Low Temperature

The ensiling process is driven by complex microbial metabolic networks, wherein dominant taxa regulate substrate degradation and fermentation progression through specific functional pathways [15,42]. Among these, carbohydrate metabolism is primarily governed by LAB, which convert water-soluble carbohydrates (WSC) into lactic and other organic acids while modulating nucleotide turnover to support ATP production. In our study, both carbohydrate and nucleotide metabolism were significantly upregulated during the late stages of ensiling, consistent with the increased abundance of LAB observed at 60 and 90 days. This indicates an active fermentation profile sustained by lactic acid-producing communities during peak microbial succession [17].

In contrast, amino acid metabolism was significantly downregulated in the OL77 group during late fermentation compared to TCP and control treatments. This downregulation was accompanied by improved crude protein retention and reduced ammonia-N production, suggesting that OL77 suppressed proteolytic activity—likely by rapidly lowering pH and inhibiting the proliferation of proteolytic bacteria. This finding echoes prior studies highlighting the regulatory role of homofermentative LAB in limiting proteolysis via acid stress [43]. Unexpectedly, membrane transport functions were more enriched in the OL77 group, despite previous reports indicating that LAB dominance typically reduces transport-related gene abundance [44]. While *L. plantarum* abundance was suppressed in OL77-inoculated silage, the enhanced membrane transport activity suggests that other taxa within the community—potentially including OL77 itself—engaged in more active exchange of metabolic intermediates. This finding underscores the multifactorial nature of microbial contributions to functional traits and highlights the need for strain-level resolution in interpreting metagenomic predictions. Additional functional categories—including energy metabolism, signal transduction, and cell growth—were upregulated in OL77-treated silage during early fermentation stages, indicating that this strain activated key physiological systems required for niche colonization under thermal stress. Most of these functions peaked at day 60 and declined thereafter, indicating that the silage system had entered a stable, low-activity maintenance phase. This trajectory reflects a classical ecological succession pattern: high activity during colonization, followed by functional saturation and community stabilization once fermentation reaches completion. Together, these findings suggest that OL77 not only restructured the microbial taxonomy, but also redirected the functional metabolism of the silage ecosystem, accelerating the transition toward anaerobic stabilization and protein preservation under cold conditions.

## 5. Conclusions

This study evaluated the impact of *Pediococcus pentosaceus* OL77 on the fermentation quality and microbial community dynamics of oat silage under low-temperature conditions. The hypothesis that OL77 would enhance fermentation, accelerate lactic acid production, and improve overall silage quality under cold conditions was confirmed. Inoculation with OL77 significantly accelerated pH reduction, increased lactic acid concentration, and reduced undesirable microbial growth, particularly inhibiting the proliferation of Enterobacter and Pseudomonas. The OL77-treated silage exhibited better preservation of DM, CP, and WSC compared to the control group. Furthermore, OL77 inoculation enhanced the overall fermentation stability, rapidly lowering pH and stabilizing microbial succession through early activation of carbohydrate metabolism, energy flux, and signal transduction pathways. These findings demonstrate that OL77 is an effective and reliable inoculant for improving oat silage fermentation in cold regions, supporting its potential for broader application in low-temperature agricultural systems.

## Figures and Tables

**Figure 1 microorganisms-13-02248-f001:**
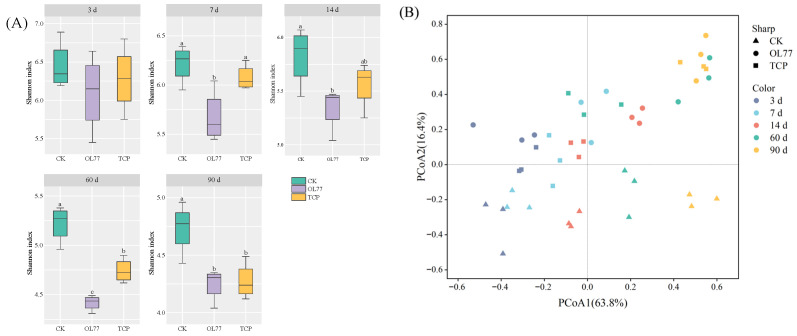
Diversity of bacterial community during oat silage. (**A**) Alpha diversity (Shannon index) of bacterial community at 3, 7, 14, 60 and 90 d of silage fermentation. Different capital letters (a–c) indicate significant differences (*p* < 0.05) among different silage groups. (**B**) Beta diversity, shape represents different inoculant treatments, and color represents different silage days.

**Figure 2 microorganisms-13-02248-f002:**
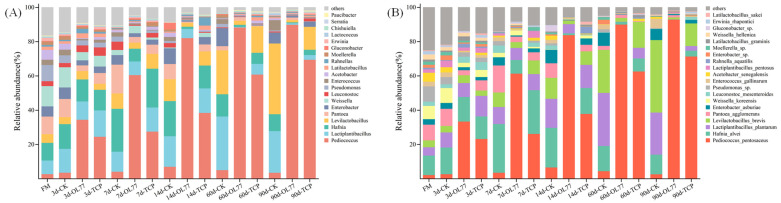
Dynamics of bacterial community composition in oat silage. (**A**) The relative abundance of the top 20 bacterial genera in oat silage under different inoculation treatments and fermentation times. (**B**) The relative abundance of the top 20 bacterial species in oat silage under different inoculation treatments and fermentation times.

**Figure 3 microorganisms-13-02248-f003:**
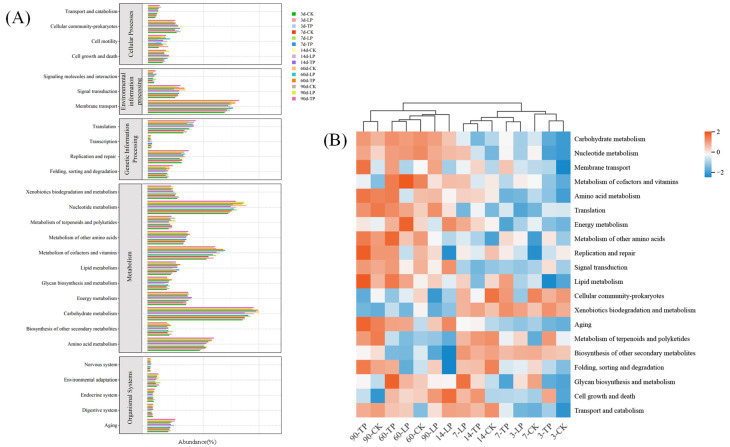
Function profiles of bacterial community. (**A**) Function profiles of bacterial community in different groups. Summary of second level of Kyoto Encyclopedia of Genes and Genomes (KEGG) orthologue functional predictions explained by PICRUSt2. (**B**) Cluster heatmap of the top 20 functions in terms of abundance proportion under different inoculant treatments and ensiling days.

**Figure 4 microorganisms-13-02248-f004:**
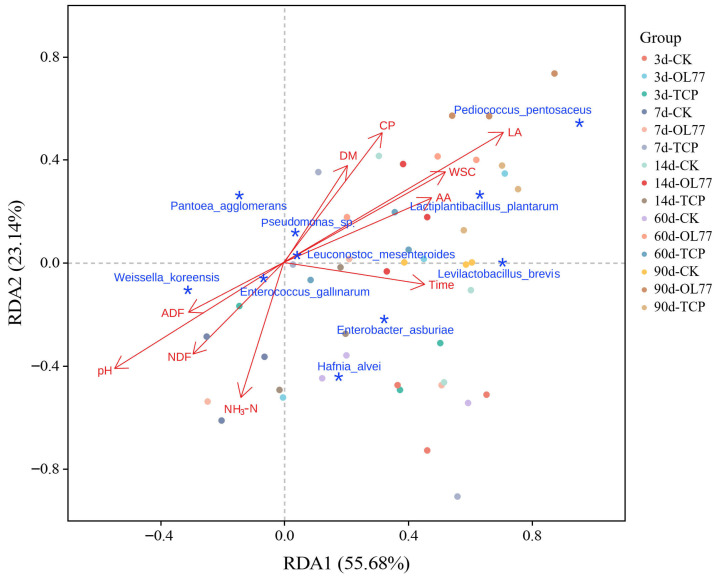
Fermentation characteristics and bacterial community redundancy analysis during oat silage process. DM, dry matter; WSC, water-soluble carbohydrate; CP, crude protein; NDF, neutral detergent fiber; ADF, acid detergent fiber; LA, lactic acid; AA, acetic acid. NH_3_-N, ammonia nitrogen; Time, silage fermentation days; *, top 10 bacterial species in relative abundance, The axes, RDA1 and RDA2, explain 55.68% and 23.14% of the total variation, respectively.

**Table 1 microorganisms-13-02248-t001:** Fermentative parameters of oat silages with different microbial inoculants at low temperature.

Items ^a^	Treatments	Ensiling Days ^b^					SEM ^c^	*p*-Value ^d^		
		3	7	14	60	90		D	T	D × T
pH	CK	6.23Aa	6.05Ab	5.71Ac	4.66Ad	4.43Ae	0.203	<0.001	<0.001	<0.001
	OL77	5.84Ca	5.17Cb	4.42Cc	3.92Cd	3.88Cd	0.209			
	TCP	6.18Aa	5.68Bb	5.03Bc	4.47Bd	4.13Be	0.209			
Lactic acid (g/kg DM)	CK	2.54Ce	4.65Cd	8.87Cc	19.29Cb	30.51Ba	2.881	<0.001	<0.001	<0.001
	OL77	9.96Ae	14.44Ad	24.18Ac	40.27Ab	45.83Aa	3.888			
	TCP	4.51Be	7.85Bd	12.23Bc	33.83Bb	41.23Aa	4.090			
Acetic acid (g/kg DM)	CK	1.37Ce	2.39Cd	4.69Bc	10.72Cb	17.32Aa	1.653	<0.001	<0.001	<0.001
	OL77	2.93Ae	3.54Ad	5.77Ac	12.07Bb	13.85Ca	1.243			
	TCP	1.68Be	2.95Bd	5.82Ac	13.70Ab	15.44Ba	1.556			
Propionic acid (g/kg DM)	CK	0.08Ce	0.35Cd	0.90Cc	2.32Cb	2.98Ca	0.313	<0.001	<0.001	<0.001
	OL77	0.25Ad	0.79Ac	2.51Ab	4.03Aa	4.06Aa	0.440			
	TCP	0.12Be	0.46Bd	1.30Bc	3.09Bb	3.73Ba	0.396			
Butyric acid (g/kg DM)	CK	ND	ND	0.06c	0.13Ab	0.35Aa	0.036	<0.001	<0.001	<0.001
	OL77	ND	ND	ND	ND	ND	0.000			
	TCP	ND	ND	ND	0.08Bb	0.14Ba	0.016			
LA/AA	CK	1.85Cab	1.95Ca	1.89Bab	1.80Cab	1.76Cb	0.025	<0.001	<0.001	<0.001
	OL77	3.40Ab	4.09Aa	4.19Aa	3.34Ab	3.31Ab	0.116			
	TCP	2.68Ba	2.67Ba	2.10Bb	2.47Bab	2.67Ba	0.070			

^a^ DM, dry matter; LA/AA, the ratio of lactic acid and acetic acid. ^b^ Different capital letters (A–C) indicate significant differences (*p* < 0.05) among different silage groups. There is a significant difference (*p* < 0.05) between the different lowercase letters (a–e) indicating the number of silage days. ^c^ SEM, standard error of the mean. ^d^ D, ensiling day; T, treatments; D × T, interaction between treatments and ensiling day.

**Table 2 microorganisms-13-02248-t002:** Nutritional composition during oat silage process.

Items ^a^	Treatments	Ensiling Days ^b^					SEM ^c^	*p*-Value ^d^		
		3	7	14	60	90		D	T	D × T
DM (g/kg FM)	CK	344.41a	342.26a	330.42b	312.44c	304.87Bd	4.424	<0.001	0.013	0.208
	OL77	344.09a	341.33a	333.92b	319.28c	313.75Ad	3.381			
	TCP	343.70a	341.08a	332.34b	316.15c	309.77Ad	3.810			
WSC (g/kg DM)	CK	138.87Aa	134.31Ab	120.58Ac	63.36Bd	47.60Ce	10.497	<0.001	0.098	<0.001
	OL77	134.35Ba	128.66Bb	109.56Bc	66.19Ad	64.68Ad	8.286			
	TCP	136.21Ba	133.20Ab	116.91Ac	67.33Ad	53.91Be	9.489			
CP (g/kg DM)	CK	102.88a	96.75Bab	94.31Bb	79.74Cc	65.52Cd	2.511	<0.001	<0.001	<0.001
	OL77	102.83a	99.50Aab	97.47Ab	92.92Ac	89.26Ad	1.390			
	TCP	103.32a	97.38Bab	95.35Bb	86.03Bc	76.18Bd	2.139			
NDF (g/kg DM)	CK	527.40Aa	524.74Ab	515.26Ac	486.47Ad	458.28Ae	7.313	<0.001	<0.001	<0.001
	OL77	523.22Ba	515.36Bb	496.52Bc	442.80Cd	437.01Ce	10.039			
	TCP	527.74Aa	522.17Ab	513.37Ac	470.38Bd	446.16Be	8.872			
ADF (g/kg DM)	CK	329.83a	327.08Aa	320.96Ab	291.76Ac	284.22Ad	5.251	<0.001	<0.001	<0.001
	OL77	327.56a	322.51Bb	303.34Cc	263.35Cd	258.28Ce	8.061			
	TCP	329.37a	326.43Aa	317.63Bb	284.72Bc	270.46Bd	6.577			
NH_3_-N (g/kg DM)	CK	4.29e	15.41Ad	49.56Ac	116.87Ab	145.17Aa	15.404	<0.001	<0.001	<0.001
	OL77	4.17e	8.38Bd	27.50Cc	73.12Cb	78.20Ca	8.737			
	TCP	4.36e	12.32Ad	37.90Bc	93.54Bb	109.84Ba	11.846			

^a^ FM, fresh matter; WSC, water-soluble carbohydrate; CP, crude protein; NDF, neutral detergent fiber; ADF, acid detergent fiber; NH_3_-N, ammonia nitrogen. ^b^ Different capital letters (A–C) indicate significant differences (*p* < 0.05) among different silage groups. There is a significant difference (*p* < 0.05) between the different lowercase letters (a–e) indicating the number of silage days. ^c^ SEM, standard error of the mean. ^d^ D, ensiling day; T, treatments; D × T, interaction between treatments and ensiling day.

## Data Availability

The original contributions presented in this study are included in the article. Further inquiries can be directed to the corresponding author.

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
