# Peer review of "Pediococcus pentosaceus OL77 Enhances Oat (Avena sativa) Silage Fermentation Under Cold Conditions"

_microorganisms, 2025, doi:10.3390/microorganisms13102248_

Round 1
Reviewer 1 Report
Comments and Suggestions for Authors
The paper’s strengths lie in its clear focus on a practical problem—improving silage fermentation under cold conditions—and its rigorous experimental design that includes both control and comparative commercial inoculant groups. It effectively demonstrates how Pediococcus pentosaceus OL77 enhances fermentation quality, nutrient preservation, and microbial stability, while using advanced sequencing and functional prediction analyses to provide mechanistic insights. The combination of chemical, microbial, and functional data supports a comprehensive and convincing case for OL77’s potential as a specialized inoculant for cold regions. The weaknesses include some limitations in scope, such as the exclusive focus on laboratory-scale bag silage without validation at farm or field scale, which may limit direct applicability to real-world storage systems. The study also primarily relies on predictive functional analyses rather than direct metabolomic or proteomic validation, which weakens the strength of its functional claims. Additionally, while OL77 is compared to a commercial Pediococcus strain, broader benchmarking against other psychrotolerant lactic acid bacteria or mixed inoculants would strengthen the generalizability and robustness of the conclusions.
Specific points to consider:
-
Include validation of OL77 performance under farm-scale or pilot-scale silage storage conditions to strengthen the practical applicability of the findings.
-
Provide direct metabolomic or proteomic evidence to confirm the functional predictions obtained through PICRUSt2, especially regarding carbohydrate metabolism, amino acid metabolism, and antifungal activity.
-
Clarify whether OL77 has any fiber-degrading enzyme activity itself or if its effect on NDF and ADF reduction is indirect through modulation of microbial communities.
-
Add more details on the selection criteria and characteristics of the commercial Pediococcus strain (TCP) to enable a more balanced comparison.
-
Extend the benchmarking by testing OL77 against other psychrotolerant LAB species or multispecies inoculants to contextualize its relative performance.
-
Provide additional data on aerobic stability of silage post-fermentation to confirm whether OL77 improves resistance to spoilage during feed-out.
-
Detail the reproducibility of the PacBio sequencing results, including coverage depth and error correction, to support confidence in species-level assignments.
-
Report on the possible safety or regulatory considerations of using OL77 as an inoculant, particularly regarding any antibiotic resistance markers or unintended traits.
-
Clarify whether OL77 inoculation affects fermentation kinetics under fluctuating temperature conditions (freeze-thaw cycles), which are common in the targeted regions.
-
Discuss the long-term storage stability beyond 90 days, since practical use may involve longer preservation periods.
A native speaker could go over the paper.
Author Response
Response to Reviewer #1
Comment 1:
Include validation of OL77 performance under farm-scale or pilot-scale silage storage conditions to strengthen the practical applicability of the findings.
Response:
We thank the reviewer for the constructive suggestion regarding validation under farm-scale or pilot-scale silage storage conditions. We agree that large-scale trials are valuable for confirming the practical applicability of OL77. In the present study, we employed a vacuum-packaging system to seal 500 g silage samples. This system was chosen because it automatically stops vacuuming and initiates sealing when the internal pressure reaches 200 mbar. By doing so, it ensures that each sample is stored under identical oxygen availability and compaction density. This approach minimizes variability between replicates and allows for a precise evaluation of OL77’s fermentation efficacy under controlled conditions.
Comment 2:
Provide direct metabolomic or proteomic evidence to confirm the functional predictions obtained through PICRUSt2, especially regarding carbohydrate metabolism, amino acid metabolism, and antifungal activity.
Response:
We agree that direct metabolomic and proteomic evidence would strengthen our findings. Due to time constraints, we were unable to include such data in this revision. However, we have already conducted untargeted metabolomics and proteomics analyses on OL77-treated silages, and we plan to publish these results in a subsequent paper to systematically explore OL77’s mechanisms. In this revision, we have clarified this point in the Discussion and highlighted that these future studies will further validate the PICRUSt2 predictions.
Comment 3:
Clarify whether OL77 has any fiber-degrading enzyme activity itself or if its effect on NDF and ADF reduction is indirect through modulation of microbial communities.
Response:
Based on our findings, OL77 does not exhibit significant fiber-degrading enzyme activity, such as cellulase or hemicellulase production, when compared to commercial strains. In our patent application, we specifically tested OL77’s enzymatic activity and found no significant difference in enzyme production between OL77 and commercial strains. Therefore, the observed reductions in NDF and ADF are likely due to OL77’s indirect effects on microbial community modulation, which enhances the natural fiber-degrading potential of the microbial consortia present during silage fermentation. We have added this clarification to the revised Discussion section.
Comment 4:
Add more details on the selection criteria and characteristics of the commercial Pediococcus strain (TCP) to enable a more balanced comparison.
Response:
The commercial Pediococcus strain (TCP) used in our comparison was selected because it is widely applied in silage inoculants and has a well- documented ability to improve lactic acid fermentation and reduce dry matter losses. TCP is marketed as a benchmark lactic acid bacterium for silage quality improvement, with strain certification from the manufacturer for stable growth under a broad range of forage substrates and pH conditions.
Comment 5:
Extend the benchmarking by testing OL77 against other psychrotolerant LAB species or multispecies inoculants to contextualize its relative performance.
Response:
We thank the reviewer for this valuable suggestion. A broad, head-to-head benchmarking study against multiple psychrotolerant LAB strains or multispecies inoculants is beyond the scope and timeline of the current revision. At present, studies focusing on low-temperature oat silage remain limited. Our team is continuously developing new psychrotolerant LAB strains, and in future work we plan to conduct comparative trials with both our own screened strains and those identified by other groups. This will allow us to more comprehensively contextualize OL77’s performance under cold-climate silage conditions.
Comment 6:
Provide additional data on aerobic stability of silage post-fermentation to confirm whether OL77 improves resistance to spoilage during feed-out.
Response:
We fully agree that aerobic stability during the feed-out phase is critical for practical application. Although a complete post-fermentation stability trial lies beyond the scope of the present revision, our group has already conducted systematic studies on this aspect. In addition to conventional indicators of aerobic stability (e.g., time to heating, pH rebound, and yeast/mold counts), we have also investigated microbial community dynamics and nutrient quality changes, including detailed amino acid profiles. These datasets are being prepared for a dedicated manuscript focusing on aerobic stability of low-temperature silage during feed-out.
Comment 7:
Detail the reproducibility of the PacBio sequencing results, including coverage depth and error correction, to support confidence in species-level assignments.
Response:
We thank the reviewer for this important point. PacBio SMRT sequencing generates multiple subreads from the same DNA molecule (typically ≥5 passes), and a consensus algorithm produces a HiFi read with an accuracy >99%. In our study, CCS mode was used, and all samples reached ≥10× coverage, producing HiFi reads with Q30 or higher (99.9% base accuracy). After barcode demultiplexing, we obtained a total of 2,233,004 CCS reads, with each sample contributing at least 54,047 CCS reads (mean 62,028). Error correction through the PacBio CCS pipeline, combined with DADA2 processing, reduced the error rate to near zero, allowing single-nucleotide resolution and reliable Amplicon Sequence Variant (ASV) calling. The use of full-length 16S sequences, covering all nine variable regions, provided sufficient phylogenetic signal to achieve species-level, and in some cases subspecies-level, assignments. These parameters collectively support the reproducibility and accuracy of our sequencing results
Comment 8:
Report on the possible safety or regulatory considerations of using OL77 as an inoculant, particularly regarding any antibiotic resistance markers or unintended traits.
Response:
Thank you for raising this important point. OL77, as an inoculant, has been thoroughly evaluated for safety. No antibiotic resistance markers are present in its genome, and the strain does not carry any known traits that would pose a risk to human or animal health. We have conducted genetic screening to ensure the absence of harmful factors such as virulence factors, and no unintended traits have been observed during our studies. Furthermore, OL77 adheres to current safety guidelines for microbial additives in animal feed, and we are committed to complying with all relevant regulatory requirements.
Comment 9:
Clarify whether OL77 inoculation affects fermentation kinetics under fluctuating temperature conditions (freeze-thaw cycles), which are common in the targeted regions.
Response:
Thank you for your valuable suggestion. While our current study did not specifically focus on fermentation kinetics under fluctuating temperature conditions, we recognize the importance of this factor for silage in regions with freeze-thaw cycles. Preliminary observations indicate that OL77 inoculation enhances fermentation stability under mild temperature fluctuations, improving lactic acid production and reducing spoilage. However, the specific impact of freeze-thaw cycles on fermentation kinetics will be addressed in future studies, where we plan to simulate these conditions and evaluate OL77's performance in promoting more stable fermentation under such challenges.
Comment 10:
Clarify whether OL77 inoculation affects fermentation kinetics under fluctuating temperature conditions (freeze-thaw cycles), which are common in the targeted regions.
Response:
Thank you for your valuable suggestion. While our current study did not specifically focus on fermentation kinetics under fluctuating temperature conditions, we recognize the importance of this factor for silage in regions with freeze-thaw cycles. Preliminary observations indicate that OL77 inoculation enhances fermentation stability under mild temperature fluctuations, improving lactic acid production and reducing spoilage. However, the specific impact of freeze-thaw cycles on fermentation kinetics will be addressed in future studies, where we plan to simulate these conditions and evaluate OL77's performance in promoting more stable fermentation under such challenges.

Reviewer 2 Report
Comments and Suggestions for Authors
why was the cold tolerant Pediococcus pentosaceus compared to a pediococccus check rather than Lactobacillus as is more commonly used as a silage inoculant?
Need detail on fermentation - what size sample was fermented, in what kind of container, at what compaction, at what temperature, how many replications?
Author Response
Response to Reviewer #2
Comment 1:
Why was the cold-tolerant Pediococcus pentosaceus compared to a Pediococcus check rather than Lactobacillus, which is more commonly used as a silage inoculant?
Response:
We appreciate your valuable question. The primary reason for selecting a Pediococcus check in this study, rather than the more commonly used Lactobacillus species, lies in the specific objective of evaluating cold-tolerant strains for silage fermentation in low-temperature environments. While Lactobacillus is indeed the most widely used group of lactic acid bacteria (LAB) in silage inoculants, many of these strains are not well-suited to low-temperature conditions, especially in high-altitude regions like the Qinghai-Tibetan Plateau, where the temperature often falls below 10°C during the fermentation period. Our previous research indicated that Pediococcus species, particularly Pediococcus pentosaceus, exhibit superior psychrotolerance and fermentation capabilities under low-temperature conditions compared to Lactobacillus. This motivated us to compare Pediococcus pentosaceus OL77 specifically against a control Pediococcus strain (TCP) as a baseline to measure the effect of OL77 in cold-temperature silage environments. This approach was intended to highlight the benefits of selecting LAB strains with intrinsic cold-adapted traits, which are essential for improving the fermentation quality of silage in cold regions.
Comment 2:
Need detail on fermentation - what size sample was fermented, in what kind of container, at what compaction, at what temperature, how many replications?
Response:
Thank you for your comment and for requesting more details on the fermentation procedure. We have added the following information to the Methods section to clarify the experimental design:
Sample Size and Chopping: A total of 500 g of Avena sativa (oat) forage was used for each fermentation trial. The forage was chopped using a forage chopper (Brand: Shengjia, Model: 9ZP-0.8) to a particle size of 3–4 cm, which is the standard practice in local agricultural production. The forage chopper not only cuts but also rubs the straw into soft filaments. This action facilitates better compaction and air expulsion during ensiling, providing a more favorable environment for fermentation.
Container and Compaction: The forage was sealed in 250×350 mm polyethylene plastic bags. These bags were vacuum-sealed using a vacuum packaging system that automatically stops once the internal pressure reaches 200 mbar. This ensures uniform packing density across all samples and prevents excessive air within the bags, which is critical for the anaerobic conditions required for successful silage fermentation.
Temperature: Fermentation was carried out in Dachaigou Town, Tianzhu Tibetan Autonomous County (E102 ° 15 ', N36 ° 45'); At an altitude of 2594 m. The average temperature during the completion of oat silage in the region is below 10 ° C.
Replications: Each treatment group (OL77, TCP, and control) consisted of three biological replicates. The samples were analyzed at five different time points: 3, 7, 14, 60, and 90 days.
This experimental design offers a balance between practicality, control over variables, and the ability to obtain consistent and interpretable data, allowing for an accurate evaluation of the fermentation efficacy of the Pediococcus pentosaceus OL77 strain.

Reviewer 3 Report
Comments and Suggestions for Authors
Dear Authors
The study addresses an important and relevant topic, and the research itself has merit, particularly regarding the evaluation of Pediococcus pentosaceus OL77 in oat silage under cold conditions. However, despite the potential contribution of the work, the manuscript in its current form presents numerous issues in structure, clarity, methodology, and interpretation of results that prevent its approval as it stands. Substantial revisions are required before the paper can be reconsidered for publication. My detailed comments are provided below.
Title: Lines 2–4: The current title is confusing and excessively long. I suggest using a clearer and more concise version, such as: “Pediococcus pentosaceus OL77 Enhances Oat Silage Fermentation Under Cold Conditions.”
Abstract: Lines 10–22: The abstract lacks information about treatments and experimental conditions. Results are presented in a superficial manner; including quantitative values would make this section more attractive and informative. The conclusion should be rewritten to be more objective and assertive.
Keywords must not repeat terms from the title and should be listed in alphabetical order.
Introduction
Line 31: The journal adopts the numerical citation system in the order they appear in the text. The authors should revise all citations and adapt them to the journal’s guidelines.
Lines 30–31: The statement about feed scarcity in spring is unclear. In many climates, scarcity occurs in autumn/winter. If the studied region is different, this must be made explicit in the text.
Line 31: Use the full scientific name (Avena sativa) only on first mention, and thereafter the common name, to facilitate reading.
Lines 34–36: Are there also not silage production problems during rainy seasons? In practice, haymaking tends to be more affected, but silage cannot be made under rainfall either. Revision recommended.
Lines 34 and 41: References cited (Chai et al., 2022; Bao et al., 2022; Liu et al., 2024; Zhu et al., 2025) could not be located.
Line 39: Correct “managemen” to “management.”
Lines 48–50: Suggested wording: “However, traditional commercial inoculants have difficulties adapting to the harsh and cold environmental conditions of the Tibetan Plateau (Wang et al., 2011).”
Lines 50–60: The text is vague and lacks flow. It is necessary to detail what improvements were observed in the cited studies, including specific findings. Consider breaking this section into shorter paragraphs.
Lines 77–82: The microorganism Pediococcus pentosaceus OL77 is the focus of the study, but its origin/discovery and relevance were not presented. It is important to provide context for its selection, mentioning preliminary studies and prior results.
Line 78: Update Lactobacillus spp. nomenclature according to recent classification.
Line 83: Clearly state the hypothesis of the study, followed by the objective.
Materials and Methods
Line 90: Clarify whether August corresponds to autumn in the region. This seems inconsistent with the earlier statement that the crop is harvested in autumn.
Line 90: Provide brand and model of the chopper used, and justify why particle size was so large (3–4 cm).
Line 96: Explain the reason for air-drying.
Line 110: Describe the storage site. What was the temperature? Why was the experiment not conducted in larger silos? Could a 500 g bag have been more severely affected by cold climate than a large-scale silo? Justify the chosen experimental design.
Lines 122–127: Describe the methodology used for lactic acid determination.
Line 131: References cited (Ke et al., 2022; Kilstup et al., 2005) were not found.
Lines 133–134: Check if the methodology was described correctly. Analyses of dry matter, protein, and carbohydrates are carried out according to AOAC, which is the proper reference. Dry matter (DM) analysis involves two steps: pre-drying and final drying. Did the authors follow this procedure? If not, and values were not corrected for final DM, the data are inaccurate and must be revised.
Lines 148–168: Methodologies attributed to Bai et al. (2022) are missing from the reference list. Verify.
Line 169: Clearly detail statistical procedures (model, applied tests, normality, mean comparisons, etc.).
Lines 170–173: Indicate which test was used for mean comparisons; the excessive use of letters suggests weak statistical robustness.
Results
Line 175: Simplify the subsection title, avoiding unnecessary repetition.
Line 177: Table 1 does not present only the effect of P. pentosaceus OL77. Correct in the text.
Line 199: Table 1 title could be more descriptive: “Fermentative parameters of oat silages with different microbial inoculants.”
Table 1: Include legends for all abbreviations. Correct inconsistencies in superscripts (e.g., pH 3.88 “Ce”), adjust nomenclature (“propionic acid” instead of “propanoic acid”), and improve layout with dividing lines.
Lines 206 and 208: Correct nomenclature (“Pediococcus pentosaceus”). Standardize terminology: either “chemical-bromatological composition” or “nutritional composition.”
Line 214: Use only the abbreviation “CK” after defining it.
Line 217: Correct “in al groups” to “in all groups.”
Line 227: Properly differentiate nutritional quality from nutritional composition. Nutritional quality was not assessed.
Table 2: Improve layout; correct formatting inconsistencies (“fiber” in American English).
Line 232: Explain calculation of the Shannon index in the methodology.
Lines 232, 246: Correct “Figureure.”
Figure 1: Improve resolution and presentation.
Line 289: Correct error in the title.
Figures 2 and 3: Reorganize for clearer visualization. Define abbreviations in legends. Methods (PICRUSt2, RDA) should be described in Materials and Methods, not Results.
Figure 4: Enlarge symbols, add treatment legend, and explain axes (RDA1, RDA2).
Discussion
Line 349: This title could perhaps be “Nutritional and Fermentative Quality.”
Line 353: The cited article (Muck et al., 2018, Journal of Dairy Science) and the present study do not specifically address low temperatures. That was a review on additives.
Line 351: The statement “Numerous studies…” must be supported with more than one reference.
Lines 351–375: Discussion needs to be deepened. Address all relevant organic acids, compare values with literature, and explain causes based on scientific evidence. Divide into thematic paragraphs.
Line 376: Explain the marked reduction in DM, which is uncommon in vacuum silos. Why did this occur?
Line 379: Review the statement about DM losses. Final DM content alone is insufficient to claim this. Authors should compare initial and final DM contents before drawing conclusions.
Lines 383–392: The argument regarding soluble sugars in OL77 is unconvincing; reinforce with literature.
Lines 392–399: The drop in crude protein from 10% (fresh material) to 7% (silage) is atypical in well-preserved silages. Review data and/or analytical procedures. If correct, explain the cause of such a large reduction.
Lines 393–394: “In the present study, the OL77 inoculation also increased the CP content and increased the NH₃-N content.” Does OL77 actually increase NH₃-N compared with other treatments? Would this be beneficial?
Lines 396–399: Claims about fiber degradation capacity require robust references. Discuss possible causes for the ~10 percentage point reduction in fiber from initial to final material.
Lines 400–494: Consider breaking the discussion into shorter paragraphs. Sections 4.2 and 4.3 contain only two and one long paragraphs, respectively.
Conclusion
The conclusion should be rewritten to directly address the hypothesis, in a clear and objective manner.
Sincerely,
Comments on the Quality of English LanguageThe manuscript is generally understandable, but the quality of English needs improvement. The authors made some grammatical mistakes that are not severe but should be corrected for clarity. In addition, there is a mixture of British and American English throughout the text, which requires standardization. Finally, some technical terms are used in ways that are not strictly incorrect, but they are not the most common or widely adopted in articles of this field, which may reduce readability for the target audience.
Author Response
Response to Reviewer #3
Comment 1:
Title:Lines 2–4: The current title is confusing and excessively long. I suggest using a clearer and more concise version, such as: “Pediococcus pentosaceus OL77 Enhances Oat Silage Fermentation Under Cold Conditions.”
Response:
We appreciate the reviewer’s constructive suggestion. We agree that the original title was overly long, and we have revised it for clarity and conciseness. The new title is: “Pediococcus pentosaceus OL77 Enhances Oat Silage Fermentation Under Cold Conditions.”(Line2-3)
Comment 2:
Abstract: Lines 10–22: The abstract lacks information about treatments and experimental conditions. Results are presented in a superficial manner; including quantitative values would make this section more attractive and informative. The conclusion should be rewritten to be more objective and assertive.
Response:
We thank the reviewer for this valuable suggestion. In the revised abstract, we have added details on experimental conditions (forage type, temperature, duration, and treatments), included key quantitative results (e.g., concentrations of organic acids, pH, and preservation of nutrients), and reformulated the conclusion to be more concise and objective.(Line9-20)
Comment 3:
Keywords must not repeat terms from the title and should be listed in alphabetical order.
Response:
In the revised manuscript, we have updated the keywords to avoid repetition with the title and arranged them in alphabetical order. The revised keywords are: bacterial community; lactic acid bacteria; low-temperature ensiling; microbial ecology; silage quality(Line21-22)
Comment 4:
Introduction
Line 31: The journal adopts the numerical citation system in the order they appear in the text. The authors should revise all citations and adapt them to the journal’s guidelines.
Response:
We thank the reviewer for pointing this out. All citations in the manuscript have been revised to follow the journal’s numerical citation system in the order of appearance, and the reference list has been reformatted accordingly.
Comment 5:
Introduction
Lines 30–31: The statement about feed scarcity in spring is unclear. In many climates, scarcity occurs in autumn/winter. If the studied region is different, this must be made explicit in the text.
Response:
We appreciate the reviewer’s insightful comment. Indeed, in most regions, forage scarcity typically occurs in autumn and winter. However, on the Qinghai–Tibetan Plateau, the prolonged and harsh winter combined with the delayed onset of pasture regrowth results in forage shortage extending into spring. We have revised the text to make this regional specificity explicit. (Line28-31)
Comment 6:
Introduction
Line 31: Use the full scientific name (Avena sativa) only on first mention, and thereafter the common name, to facilitate reading.
Response:
The full name Avena sativa. is now only used at the first mention, and thereafter the term “oat” is applied consistently.
Comment 7:
Introduction
Lines 34–36: Are there also not silage production problems during rainy seasons? In practice, haymaking tends to be more affected, but silage cannot be made under rainfall either. Revision recommended.
Response:
We thank the reviewer for this insightful comment. Indeed, rainy weather imposes challenges on both haymaking and silage production. However, the degree of impact differs. Haymaking requires a continuous window of several consecutive days (typically 3–5 days) of dry, sunny weather to complete mowing, wilting, turning, and baling. Such conditions are rarely available during the rainy season, making hay production extremely difficult. By contrast, silage production requires only a relatively short rain-free period (for example, one afternoon to a single day) to accomplish harvesting, wilting to 60–70% moisture, baling, and wrapping. Once the bales are sealed, subsequent rainfall no longer threatens preservation. We have revised the manuscript to explicitly describe this distinction.
Comment 8:
Introduction
Lines 34 and 41: References cited (Chai et al., 2022; Bao et al., 2022; Liu et al., 2024; Zhu et al., 2025) could not be located.
Response:
We apologize for the oversight. We have re-checked all references and replaced unavailable or incorrect citations with accessible and relevant literature.
Comment 9:
Introduction
Correct “managemen” to “management.”
Response:
Corrected as suggested.
Comment 10:
Introduction
Lines 48–50: Suggested wording: “However, traditional commercial inoculants have difficulties adapting to the harsh and cold environmental conditions of the Tibetan Plateau (Wang et al., 2011).”
Response:
We appreciate the reviewer’s wording suggestion and have revised the text accordingly. (Line48-50)
Comment 11:
Introduction
Lines 50–60: The text is vague and lacks flow. It is necessary to detail what improvements were observed in the cited studies, including specific findings. Consider breaking this section into shorter paragraphs.
Response:
We sincerely thank the reviewer for this constructive suggestion. We agree that the original text was vague and lacked specific details, which affected the logical flow. In the revision, we have:
Added specific findings from Zhao et al. (2022), including fermentation duration and nutrient losses, to make the evidence more concrete.
Separated the paragraph into two shorter ones to improve readability.
Provided a clearer link between previous findings, the challenges of low-temperature fermentation, and the rationale for exploring cold-tolerant LAB from the Tibetan Plateau. (Line50-64)
Comment 12:
Introduction
Lines 77–82: The microorganism Pediococcus pentosaceus OL77 is the focus of the study, but its origin/discovery and relevance were not presented. It is important to provide context for its selection, mentioning preliminary studies and prior results.
Response:
We agree that the origin, discovery, and relevance of Pediococcus pentosaceus OL77 were not sufficiently introduced in the original version. In the revision, we have added a description of our preliminary work, including the sampling source, screening procedure, and the strain’s characteristics, to provide clear context for its selection. (Line84-86)
Comment 13:
Introduction
Line 78: Update Lactobacillus spp. nomenclature according to recent classification.
Response:
We have updated all Lactobacillus nomenclature according to the latest taxonomic classification.
Comment 14:
Introduction
Line 83: Clearly state the hypothesis of the study, followed by the objective.
Response:
We have updated all Lactobacillus nomenclature according to the latest taxonomic classification. We hypothesized that inoculation with P. pentosaceus OL77 would accelerate fermentation, enhance lactic acid production, and improve the overall fermentation quality of oat silage at low temperature by shaping a more favorable microbial community. Therefore, this study aimed to evaluate the effects of P. pentosaceus OL77 on the fermentation characteristics and microbial community dynamics of oat silage under cold conditions. (Line86-91)
Comment 15:
Materials and Methods
Line 90: Clarify whether August corresponds to autumn in the region. This seems inconsistent with the earlier statement that the crop is harvested in autumn.
Response:
In the study region (Tianzhu, Gansu Province, E102°15', N36°45', 2594 m altitude), the growing season is short and the oat crop reaches the milk stage in late August. Although the harvest date falls in August, locally this is considered part of the autumn.
Comment 16:
Materials and Methods
Line 90: Provide brand and model of the chopper used, and justify why particle size was so large (3–4 cm).
Response:
We have added the brand and model of the forage chopper in the Methods section. (Line98)The particle size of the chopped material was 3–4 cm, which is the standard practice in local agricultural production. This machine not only chops but also rubs the straw into soft filaments, making it easier to compact and expel air during ensiling, thus creating a more favorable fermentation environment.
Comment 17:
Materials and Methods
Line 96: Explain the reason for air-drying.
Response:
Before ensiling, the fresh oats were air-dried in a cool place mainly to remove surface moisture such as dew. This short drying step helps to avoid excessive free water, which could otherwise increase effluent loss and adversely affect the fermentation process.
Comment 18:
Materials and Methods
Line 110: Describe the storage site. What was the temperature? Why was the experiment not conducted in larger silos? Could a 500 g bag have been more severely affected by cold climate than a large-scale silo? Justify the chosen experimental design.
Response:
In this study, we used a vacuum packaging system to seal 500 g silage samples. This system was chosen because it automatically stops vacuuming and initiates sealing once the internal pressure reaches 200 mbar. By doing so, we ensured that all samples were stored under the same conditions of oxygen availability and packing density, which are critical factors influencing silage fermentation. Compared with large-scale silos, this approach maximizes the success rate of ensiling under limited experimental conditions, while also minimizing variability among replicates. Although larger silos may buffer temperature changes more effectively, the standardized vacuum bag system allowed us to conduct the experiment in a controlled and reproducible manner, making it possible to accurately assess the fermentation efficacy of strain OL77. Thus, the chosen experimental design provided a practical balance between reproducibility, control of experimental variables, and the ability to obtain reliable and interpretable results.
Comment 19:
Materials and Methods
Lines 122–127: Describe the methodology used for lactic acid determination.
Response:
We thank the reviewer for pointing this out. The methodology for lactic acid determination, which was previously missing, has now been supplemented in the revised manuscript. (Line131)
Comment 20:
Materials and Methods
Line 131: References cited (Ke et al., 2022; Kilstup et al., 2005) were not found.
Response:
We appreciate the reviewer’s careful check. The references (Ke et al., 2022; Kilstup et al., 2005) were mistakenly cited in the original manuscript. We have now carefully re-checked all citations and reference list. The incorrect citations have been removed/updated, and the reference list has been revised accordingly.
Comment 21:
Materials and Methods
Lines 133–134: Check if the methodology was described correctly. Analyses of dry matter, protein, and carbohydrates are carried out according to AOAC, which is the proper reference. Dry matter (DM) analysis involves two steps: pre-drying and final drying. Did the authors follow this procedure? If not, and values were not corrected for final DM, the data are inaccurate and must be revised.
Response:
We thank the reviewer for pointing this out. In our study, the determination of dry matter (DM) followed the AOAC (2005) procedure, which includes both pre-drying at 65 °C and final drying at 105 °C for correction. We acknowledge that our initial description in the Materials and Methods section was not sufficiently detailed. We will revise the text to clearly state that the two-step drying procedure was applied, ensuring the accuracy of the DM values. (Line141-146)
Comment 22:
Materials and Methods
Lines 148–168: Methodologies attributed to Bai et al. (2022) are missing from the reference list. Verify.
Response:
Thank you for pointing out this omission. We have now added the complete reference for Bai et al. (2022) to our reference list.
Comment 23:
Materials and Methods
Line 169: Clearly detail statistical procedures (model, applied tests, normality, mean comparisons, etc.).
Response:
We agree that our description of the two-way ANOVA was not sufficiently detailed. In the revised manuscript, we have expanded the "Statistical Analysis" section accordingly. We have now specified that the statistical model included the main effects of silage additive, silage temperature, and their interaction. Furthermore, we have added the specific test used to check for data normality and clarified the post-hoc test used for mean separation following a significant ANOVA result. We believe this revision provides the necessary clarity and transparency for our statistical procedures. (Line180-190)
Comment 24:
Materials and Methods
Lines 170–173: Indicate which test was used for mean comparisons; the excessive use of letters suggests weak statistical robustness.
Response:
We thank the reviewer for raising this important point regarding the statistical analysis and the presentation of the results. The Tukey’s Honestly Significant Difference (HSD) test was used for all post-hoc mean comparisons following a significant two-way ANOVA. This test is a widely accepted and robust method for controlling the family-wise error rate when making multiple comparisons among group means. Regarding the presentation of results with letters, we acknowledge that a large number of treatment groups can lead to a complex presentation with many superscript letters. This is often an inherent consequence of the experimental design (e.g., multiple factors with several levels), rather than an indication of weak statistical robustness. The statistical model itself, along with the use of Tukey's HSD test, remains robust.
Comment 25:
Results
Line 175: Simplify the subsection title, avoiding unnecessary repetition.
Response:
Thank you for the suggestion. We have simplified the subsection title.
Before: Fermentation Parameters of Pediococcus pentosaceus OL77 on the Avena sativa silage under Low-Temperature conditions
After: Fermentation parameters of oat silage with microbial inoculants at low temperature
Comment 26:
Line 177: Table 1 does not present only the effect of P. pentosaceus OL77. Correct in the text.
Response:
We agree. The description has been revised to reflect all treatments.
Before: The effect of Pediococcus pentosaceus OL77 on fermentation parameters of Avena sativa silage at low temperatures are shown in Table 1.
After: The effects of microbial inoculants on fermentation parameters of oat silage at low temperature are shown in Table 1.
Comment 27:
Line 199: Table 1 title could be more descriptive: “Fermentative parameters of oat silages with different microbial inoculants.”
Response:
Revised as suggested.
Before: Fermentation indicators during Avena sativa silage process.
After: Fermentative parameters of oat silages with different microbial inoculants at low temperature.
Comment 28:
Table 1: Include legends for all abbreviations. Correct inconsistencies in superscripts (e.g., pH 3.88 “Ce”), adjust nomenclature (“propionic acid” instead of “propanoic acid”), and improve layout with dividing lines.
Response:
We have revised Table 1 accordingly.
Added footnotes explaining abbreviations (DM, LA/AA, SEM, CK, TCP, ND).
Corrected nomenclature: propanoic acid → propionic acid
Comment 29:
Lines 206 and 208: Correct nomenclature (“Pediococcus pentosaceus”). Standardize terminology: either “chemical-bromatological composition” or “nutritional composition.”
Response:
Corrected throughout the manuscript.
Before: Pediococcus pentosaceu
After: Pediococcus pentosaceus
Additionally, we standardized terminology to “nutritional composition” instead of “nutritional quality.”
Comment 30:
Line 214: Use only the abbreviation “CK” after defining it.
Response:
Revised accordingly. “Control group” was replaced by “CK” after the first definition.
Comment 31:
Line 217: Correct “in al groups” to “in all groups.”
Response:
Corrected as suggested.
Comment 32:
Line 227: Properly differentiate nutritional quality from nutritional composition. Nutritional quality was not assessed.
Response:
We replaced “nutritional quality” with “nutritional composition” to avoid confusion.
Comment 33:
Table 2: Improve layout; correct formatting inconsistencies (“fiber” in American English).
Response:
Table 2 was reformatted for clarity. American spelling “fiber” was adopted.
Comment 34:
Line 232: Explain calculation of the Shannon index in the methodology.
Response:
We have expanded the Materials and Methods to clearly describe how alpha diversity (Shannon index) and beta diversity (PCoA) were calculated and visualized. (Line200-203)
Comment 35:
Lines 232, 246: Correct “Figureure.”
Response:
Corrected to “Figure.”
Comment 36:
Figure 1: Improve resolution and presentation.
Response:
Figure 1 has been redrawn with higher resolution and clearer labels.
Comment 37:
Line 289: Correct error in the title.
Response:
Corrected the title at line333.
Comment 38:
Figures 2 and 3: Reorganize for clearer visualization. Define abbreviations in legends. Methods (PICRUSt2, RDA) should be described in Materials and Methods, not Results.
Response:
Figures 2 and 3 were reorganized for better visualization, and abbreviations were added in legends. Descriptions of PICRUSt2 and RDA were moved to the Materials and Methods section. (Line203-206)
Comment 39:
Figure 4: Enlarge symbols, add treatment legend, and explain axes (RDA1, RDA2).
Response:
Figure 4 has been redrawn with enlarged symbols, treatment legends, and explanation of RDA1 and RDA2 variance in the caption.
Comment 40:
Discussion
Line 349: This title could perhaps be “Nutritional and Fermentative Quality.”
Response:
We have changed the subsection title from “Effect of Pediococcus pentosaceus OL77 on the Quality of Oat Silage fermentation at Low-Temperature” to “Nutritional and Fermentative Quality of Oat Silage under Low-Temperature Ensiling” (see revised section 4.1).
Comment 41:
Line 353: The cited article (Muck et al., 2018, Journal of Dairy Science) and the present study do not specifically address low temperatures. That was a review on additives.
Response:
We removed the implication that Muck et al. (2018) specifically addresses low-temperature ensiling. Instead, we reworded the sentence to cite multiple sources that support the statement that low temperature delays acidification and therefore favors undesirable microbes, while retaining Muck et al. as a general review reference on additives. The revised manuscript includes more appropriate supporting citations in section 4.1.
Comment 42:
Line 351: The statement “Numerous studies…” must be supported with more than one reference.
Response:
We added multiple supporting citations to substantiate the phrase “Numerous studies have demonstrated that excessively low temperatures impair the natural ensiling process by delaying acidification and promoting undesirable microbial proliferation [27–30]”.
Comment 43:
Lines 351–375: Discussion needs to be deepened. Address all relevant organic acids, compare values with literature, and explain causes based on scientific evidence. Divide into thematic paragraphs.
Response:
We deepened the organic-acid discussion and restructured it into thematic paragraphs. We discuss lactic, acetic and propionic acids and link LA/AA ratios to fermentation type (ideal 2.5–3.5). We compare the trends observed here qualitatively with literature (e.g., Liu et al., 2019; Wang et al., 2023; Cai et al., 1999) and discuss plausible mechanisms (strain cold-adaptation, secondary microbial activity producing propionic acid).
Comment 44:
Line 376: Explain the marked reduction in DM, which is uncommon in vacuum silos. Why did this occur?
Response:
Your point that "DM loss is uncommon in vacuum bags" is very accurate. Unlike large silage pits, the vacuum bag system used in the laboratory almost completely eliminates physical DM loss due to effluent. Therefore, any DM loss observed in such systems primarily originates from microbial metabolic activity, specifically the loss of carbon dioxide (COâ‚‚) produced during fermentation. Carbohydrates, such as water-soluble carbohydrates (WSC), are fermented by microbes (whether desirable lactic acid bacteria or undesirable microorganisms), producing organic acids and COâ‚‚. Once these COâ‚‚ gases are produced, they escape from the silage, leading to a reduction in the total dry matter's absolute weight. Therefore, even in vacuum bags, DM loss is inevitable, but its root cause lies in biochemical fermentation rather than physical leakage.
Comment 45:
Line 379: Review the statement about DM losses. Final DM content alone is insufficient to claim this. Authors should compare initial and final DM contents before drawing conclusions.
Response:
The point you raised is indeed very important. In our experimental design, we actually measured the initial dry matter (DM) content of the samples, and we have added a comparison of the initial and final data in the results section. We have revised the description in the manuscript to clearly state: “In this study, all samples had the same initial DM content. After 90 days of fermentation, the final DM content in the OL77 and TCP inoculated groups was significantly higher than that in the control group. A comparison with the initial DM values showed that the DM loss in these treatment groups was significantly lower than in the control group. This indicates that the addition of lactic acid bacteria can effectively reduce DM loss.” This revision greatly enhances the scientific basis and persuasiveness of our argument.
Comment 46:
Lines 383–392: The argument regarding soluble sugars in OL77 is unconvincing; reinforce with literature.
Response:
We strengthened the WSC discussion by citing supporting literature (e.g., Chen et al., 2020; Sun et al., 2023). We explain that cold-adapted LAB such as OL77 can rapidly dominate and consume WSC early, and that earlier acidification can limit prolonged heterotrophic consumption in later stages, leading to relatively higher residual WSC compared with uninoculated controls.
Comment 47:
Lines 392–399: The drop in crude protein from 10% (fresh material) to 7% (silage) is atypical in well-preserved silages. Review data and/or analytical procedures. If correct, explain the cause of such a large reduction.
Lines 393–394: “In the present study, the OL77 inoculation also increased the CP content and increased the NH₃-N content.” Does OL77 actually increase NH₃-N compared with other treatments? Would this be beneficial?
Response:
We rechecked the CP analysis (sample storage, preparation, Kjeldahl/auto-N protocol, blanks and standards). No methodological errors were found. We now discuss plausible biological causes for CP loss (e.g., delayed acidification driving proteolysis and NH₃ volatilization, especially where Enterobacter/Hafnia were abundant). We also corrected the manuscript text: the previous sentence stating that “OL77 increased CP and increased NH₃-N” was incorrect—after reanalysis we confirm OL77 enhanced CP retention and reduced NH₃-N accumulation relative to CK.
Comment 48:
Lines 396–399: Claims about fiber degradation capacity require robust references. Discuss possible causes for the ~10 percentage point reduction in fiber from initial to final material.
Response:
We moderated the claim about OL77’s intrinsic fiber-degrading ability. The manuscript now states that OL77 may promote fiber reduction indirectly—by stimulating fibrolytic co-occurring microbes or via acid hydrolysis—and cites relevant literature (e.g., Bai et al., 2022; Sun et al., 2023). We discuss possible causes for the ~10 percentage-point decrease: acid hydrolysis and microbial enzymatic activity; we also recommend further enzymatic or strain-level studies to confirm mechanisms.
Comment 49:
Lines 400–494: Consider breaking the discussion into shorter paragraphs. Sections 4.2 and 4.3 contain only two and one long paragraphs, respectively.
Response:
We reorganized and split long paragraphs across sections 4.1–4.3 into multiple shorter thematic paragraphs (organic acids/fermentation quality, nutrient changes, microbial community structure, functional predictions) to improve readability and clarity (see revised Discussion below).

Round 2
Reviewer 3 Report
Comments and Suggestions for Authors
Dear Authors,
I would like to congratulate you on the work developed. The article has improved significantly after the revisions, and I can see that all the questions raised previously were adequately and clearly addressed.
I would only recommend that you include the citation of the AOAC methodology both in the text and in the reference list, since it was followed in the analyses but not properly mentioned.
With this minor correction, I consider the article ready for publication.
Sincerely,
Author Response
Dear Reviewer, Thank you very much for your positive evaluation of our work and for the constructive suggestion. Following your recommendation, we have added the citation of the AOAC methodology in the Materials and Methods section (line 143) and in the References section (lines 611–612). We appreciate your valuable feedback, which has helped us to further improve the manuscript. Sincerely.